# Synthesis, Crystal Structure, and Luminescence of Cadmium(II) and Silver(I) Coordination Polymers Based on 1,3-Bis(1,2,4-triazol-1-yl)adamantane

**DOI:** 10.3390/molecules26175400

**Published:** 2021-09-05

**Authors:** Roman D. Marchenko, Taisiya S. Sukhikh, Alexey A. Ryadun, Andrei S. Potapov

**Affiliations:** 1Kizhner Research Center, National Research Tomsk Polytechnic University, 30 Lenin Ave., 634050 Tomsk, Russia; rdm1@tpu.ru; 2Nikolaev Institute of Inorganic Chemistry, Siberian Branch of the Russian Academy of Sciences, 3 Lavrentiev Ave., 630090 Novosibirsk, Russia; sukhikh@niic.nsc.ru (T.S.S.); ryadunalexey@mail.ru (A.A.R.)

**Keywords:** adamantane, 1,2,4-triazole, silver, cadmium, coordination polymer, crystal structure, luminescence, thermal analysis, 1,3-bis(1,2,4-triazol-1-yl)adamantane

## Abstract

Coordination polymers with a new rigid ligand 1,3-bis(1,2,4-triazol-1-yl)adamantane (L) were prepared by its reaction with cadmium(II) or silver(I) nitrates. Crystal structure of the coordination polymers was determined using single-crystal X-ray diffraction analysis. Silver formed two-dimensional coordination polymer [Ag(L)NO_3_]*_n_*, in which metal ions are linked by 1,3-bis(1,2,4-triazol-1-yl)adamantane ligands, coordinated by nitrogen atoms at positions 2 and 4 of 1,2,4-triazole rings. Layers of the coordination polymer consist of rare 18- and 30-membered {Ag_2_L_2_} and {Ag_4_L_4_} metallocycles. Cadmium(II) nitrate formed two kinds of one-dimensional coordination polymers depending on the metal-to-ligand ratio used in the synthesis. Coordination polymer [Cd(L)_2_(NO_3_)_2_]*_n_* was obtained in case of a 1:2 M:L ratio, while for M:L = 2:1 product {[Cd(L)(NO_3_)_2_(CH_3_OH)]·0.5CH_3_OH}*_n_* was isolated. All coordination polymers demonstrated ligand-centered emission near 450 nm upon excitation at 370 nm.

## 1. Introduction

Since its discovery adamantane has gained a wide spectrum of applications—from polymer design to biological activity. As a rigid building unit, mono-, di-, tri-, and tetra-substituted adamantanes are used to construct covalent organic frameworks, hydrogen bounded networks, and metal-organic frameworks [1,2]. Similar to aromatic ligands, adamantane retains the rigidity, but it exhibits additional benefits: its tetrahedral structure offers access to unique node geometry, the non-polar backbone introduces hydrophobicity, and the conformational stability provides a pool of multiple ligand geometries in solution [1]. Adamantane derivatives are promising materials for porous recyclable catalytically active metal-containing polymers, in which they prevent metal leaching, polymer matrix loss, intercalation, and porosity deterioration [2].

Natural and synthetic adamantanes possess a wide range of biological activity. Amantadine, memantine, rimantadine, tromantadine, vildagliptin, saxagliptin, and adapalen are used in worldwide clinical practice for treatment of viral infections, acne, neurodegenerative disorders, and type 2 diabetes [3,4,5,6]. Amantadine and memantine demonstrated significant benefits in preventing and treating COVID-19 [7].

1,2,4-Triazole is a versatile nitrogen-donor ligand capable of forming strong coordination bonds with various metal ions [8,9]. Derivatives of 1,2,4-triazole are known as anticancer, antiproliferative, anti-inflammatory, anticonvulsant, analgesic, antioxidant agents, and possess activity against wide, but specific spectrum of living organisms [10]. 1,2,4-Triazole cycles and rigid adamantane structures can be combined in bis(1,2,4-triazolyl)adamantane ligands, which could be useful *N*,*N*’-linkers for constructing coordination polymers. Coordination polymers and metal-organic frameworks in particular are promising materials with a wide range of potential applications—gas storage and separation [11,12], enantioselective catalysis [13,14], supramolecular systems design [15], pollutant sensing [16,17], water purification [18], fabrication of electrochemical devices [19,20], and biological activity [21]. Coordination polymers based on 1,3-bis(1,2,4-triazol-4-yl)adamantane with Mo(IV), Cu(II), Zn(II), Cd(II), Ag(I), bimetallic Ag(I)/V(V), and Cu(II)/Mo(IV) were successfully synthesized and characterized by Domasevitch research group [22,23,24,25,26,27,28,29,30,31], but the coordination chemistry of its isomer, 1,3-bis(1,2,4-triazol-1-yl)adamantane remained unexplored.

Following interest in the coordination chemistry of heterocyclic adamantane derivatives [32,33,34], we have prepared a new ligand 1,3-bis(1,2,4-triazol-1-yl)adamantane (**L**) and explored its reactions with Cd(II) and Ag(I) ions. As a result, three new Ag(I) and Cd(II) coordination polymers were synthesized and characterized.

## 2. Results and Discussion

### 2.1. Synthesis of the Coordination Polymers

Silver coordination polymer [Ag(**L**)NO_3_]*_n_* (**1**) was synthesized from silver nitrate(I) and **L** in methanol at room temperature (Scheme 1). Molar ratio Ag:**L** = 1:1 was used, but the same product formed with Ag:**L** = 1:2 or 2:1.

Coordination polymers [Cd(**L**)_2_(NO_3_)_2_]*_n_* (**2**) and {[Cd(**L**)(NO_3_)_2_(CH_3_OH)]·0.5CH_3_OH}*_n_* (**3**) were obtained by the reaction of cadmium(II) nitrate and **L** in methanol solution at solvothermal conditions (80 °C). The molar ratio of reagents was Cd:L = 1:2 for **2** and 2:1 for **3**. With the ratio Cd:**L** = 1:1 the mixture of crystals of **2** and **3** was obtained.

### 2.2. Crystal Structures of the Coordination Polymers

Coordination polymer **1** crystallizes in monoclinic crystal system, *P*2_1_/*c* space group. The elementary unit consists of one Ag(I) ion, one **L** ligand, and one nitrate ion. Each silver(I) ion coordinates three **L** ligands—two of them via nitrogen atoms at positions 4 of 1,2,4-triazole rings and one ligand via nitrogen atoms at position 2 (Figure 1). Bridging **L** ligands and Ag(I) ions forms 2D layers parallel to the (-102) crystallographic plane (Figure 2). The layers are composed from two types of metallocycles—18-membered {Ag_2_L_2_} cycles and 30-membered {Ag_4_L_4_} cycles. Nitrate ions are disordered over two positions (Appendix A), in one of them (occupancy 0.53), a bridging coordination mode is observed with Ag–O distances of 2.640(2) and 2.801(2) Å. For the second position (occupancy 0.47) the coordination mode should be considered as monodentate with the shortest Ag–O distance of 2.852(3) Å. The observed coordination modes of the nitrate ions are consistent with Kleywegt’s geometrical criteria calculated from interatomic distances and angles (Appendix A) [35]. The bridging nitrate ions join the adjacent layers into a 3D structure (Appendix A).

Compound **2** is a 1D coordination polymer, the asymmetric unit consists of one Cd(II) ion, two nitrate ions, and two **L** ligands (Figure 3). Cd(II) ions are in a seven-coordinated environment, which can be best described as distorted pentagonal bipyramidal, according to calculated minimal distortion paths listed in Appendix A. Two equatorial coordination places are occupied by nitrogen atoms at position 4 of triazole rings and the remaining three by oxygen atoms of nitrate ions (Appendix A). According to Kleywegt’s criteria (Appendix A), one of the nitrate ions is coordinated in a monodentate fashion (Cd–O distance 2.388(2) Å), while the other is bidentate with Cd–O distances 2.524(2) and 2.627(2) Å.

Two Cd(II) ions are linked by two ligands into 1D chains {Cd(NO_3_)_2_(L)_2_}*_n_* oriented along crystallographic axis *a* (Figure 4). Pairs of neighboring chains participate in CH···O close contacts (2.508 and 2.598 Å) of the nitrate ions and adamantane methyne groups (Appendix A). 

Compound **3** crystallizes in a monoclinic P2_1_/c space group, the asymmetric unit contains two crystallographically independent Cd(II) cations, four nitrate ions, two ligand L molecules, and three CH_3_OH molecules (two of which are coordinated to Cd^2+^, Figure 5). Cd(II) ions are seven-coordinated and their coordination polyhedra resemble a distorted pentagonal bipyramid (Appendix A, Appendix A). The equatorial coordination places are occupied by four oxygen atoms of the nitrate ions, and one position 4 nitrogen atom of 1,2,4-triazole ring. The axial positions are occupied by a 1,2,4-triazole nitrogen atom of another L molecule and an oxygen atom of a coordinated methanol molecule. Both of the nitrate ions coordinated to each of independent Cd(II) ions can be considered as bidentate with the shortest and the longest N–O distances of 2.330(7) Å and 2.602(7) Å (Appendix A). Cd(II) cations are linked by bridging L molecules into 1D chains {Cd(NO_3_)_2_(L)(CH_3_OH)} oriented along crystallographic axis *b* (Figure 6). The chains are involved in intermolecular CH···O close contacts between the nitrate ions and 1,2,4-triazole hydrogen atoms at position 5 (2.699 Å) or between the methanol molecules and the 1,2,4-triazole hydrogen atoms (2.694 Å, Figure 6). These contacts join the chains into supramolecular 2D layers oriented parallel to *ab* plane (Appendix A).

### 2.3. Powder X-ray Diffraction, Thermal Analysis, and FT-IR Spectroscopy

The phase and chemical purity of compounds **1**–**3** were confirmed by powder X-ray diffraction (powder XRD, Appendix A) and elemental (CHN) analyses.

Thermal stability of the complexes was evaluated by thermogravimetric analysis (TGA) coupled with differential scanning calorimetry (DSC). Coordination polymer **1** shows only a slight weight loss up to 285 °C, after which it undergoes a rapid decomposition in the range of 285–325 °C, followed by a graduate weight loss (Appendix A). The residual mass at 700 °C (24%) corresponds well to silver content in compound **1** (24.5%), indicative of complete loss of nitrate ions and decomposition of the organic ligand.

Coordination polymer **2** is stable up to 290 °C, after which a decomposition process similar to the one observed for compound **1** takes place (Appendix A). Thermal behavior of the coordination polymer **3** is different because of the presence of solvate CH_3_OH molecules. It demonstrates an early weight loss of about 4% at 35–75 °C, attributable to removal of uncoordinated methanol molecules (calculated weight loss 3%). The formed product is stable in the range of 75–305 °C, after which it decomposes in two steps—rapid at 305–345 °C (55% weight loss) and gradual up to 685 °C, both attributable to loss of coordinated methanol, and decomposition of the organic ligand and cadmium nitrate (Appendix A).

Compounds **1**–**3** were studied by FT-IR spectroscopy in the solid state (Appendix A). Bands in the IR spectra of the coordination polymers **1**–**3** associated with 1,2,4-triazole ring vibrations in the range 1507–1517 cm^−1^ are shifted to higher frequencies compared to the spectrum of the free **L** ligand consistent with the participation of the heterocyclic rings in coordination.

Nitrate ion vibration bands in the spectrum of coordination polymer **1** were found near 1438 and 1337 cm^−1^, the separation of 101 cm^−1^ between them is consistent with the monodentate coordination of the nitrate ions. FT-IR spectra of compounds 2 and 3 demonstrate different patterns in the region of NO_3_^-^ vibrations. Thus, bands of asymmetric and symmetric NO_3_^-^ stretching vibrations are found at 1436 and 1283 cm^−1^ for compound **2** and 1448 and 1276 cm^−1^ for compound **3** (Appendix A). Higher separation between these bands (172 cm^−1^ versus 153 cm^−1^) is consistent with a more bidentate character of coordination of nitrate ions in compound **3**.

### 2.4. Luminescent Properties of the Coordination Polymers

The luminescent properties of the ligand **L** and the coordination polymers **1**–**3** were studied for polycrystalline samples in the solid state. The excitation and emission spectra of the free ligand contain a single band near 350 nm and 410 nm associated with π–π* transitions in 1,2,4-triazole rings [36] (Figure 7a). The excitation and emission spectra of the coordination polymers 1–3 are very similar, consistent with the intraligand nature of the luminescence. In contrast to the spectrum of the free ligand **L**, two excitation bands were detected in the spectra of the coordination polymers **1**–**3** (Figure 7b,c), the lower energy band demonstrates a bathochromic shift to 370 nm relatively to the free ligand, in addition, high-energy bands appear at 270 nm, associated with *n*–π* transitions in 1,2,4-triazole rings. The emission band also undergoes a bathochromic shift of 40 nm upon excitation at 370 nm. Excitation at 270 nm leads to the appearance of a higher-energy shoulder near 400 nm, probably associated with the π*–*n* emission pathway (Figure 7b). Similar photophysical behavior was observed previously for cadmium complexes with structurally similar 1-(1,2,4-triazol-1-yl)adamantane [32]. 

## 3. Materials and Methods

### 3.1. Synthesis of the Coordination Polymers

All reagents were of reagent grade and used as received without further purification. 1,3-Bis(1,2,4-triazol-1-yl)adamantane (**L**) was synthesized according to a previously reported procedure [37].

**[Ag(L)(NO_3_)]*_n_*** (**1**). Solution of AgNO_3_ (51 mg, 0.3 mmol) in methanol (2 mL) was added dropwise with stirring to the solution of **L** (81 mg, 0.3 mmol) in methanol (1 mL). The resulting solution was protected from light and left at room temperature without stirring. The immediately formed colorless precipitate was collected by filtration after 12 h, washed with methanol, and dried to give complex **1** in 77% yield. Anal. Calcd. (%) for C_14_H_18_AgN_7_O_3_: C 38.20, H 4.12, N 22.27; Found (%): C 38.48, H 4.02, N 22.58. FT-IR (ν, cm^−1^): 3119 (w), 2932 (w), 2860 (w), 1507 (m), 1438 (w), 1337 (s), 1274 (s), 1139 (s), 1018 (m), 974 (m), 910 (w), 826 (w), 739 (m), 657 (s). Single crystals of compound **1** as thin needles were obtained by layering an aqueous solution of AgNO_3_ over chloroform solution of L in a narrow test tube.

**[Cd(L)_2_(NO_3_)_2_]*_n_*** (**2**). Solution of Cd(NO_3_)_2_·4H_2_O (6,1 mg, 0.02 mmol) in methanol (1 mL) was added dropwise with stirring to **L** (10.8 mg, 0.04 mmol) in methanol (1 mL) in a screw-capped glass vial. The resulting solution was put in the oven and heated at 80 °C for 24 h. Colorless crystals were obtained, collected by filtration, washed with methanol, and dried to give complex **2** in 49% yield. Anal. Calcd. (%) for C_28_H_36_CdN_14_O_6_: C 43.28, H 4.67, N 25.23; Found (%): C 43.54, H 4.39, N 25.49. FT-IR (ν, cm^−1^): 3138 (w), 2943 (w), 2861 (w), 1516 (m), 1437 (s), 1284 (s), 1129 (s), 1035 (s), 984 (s), 858 (m), 817 (m), 739 (m), 660 (s).

**[Cd(L)(NO_3_)_2_(CH_3_OH)]·0.5CH_3_OH** (**3**). Cd(NO_3_)_2_·4H_2_O (12.3 mg, 0.04 mmol) in methanol (1 mL) was added dropwise with stirring to **L** (5.4 mg, 0.02 mmol) in methanol (1 mL). Then the same process was used as for **2** to give complex **3** in 40% yield. Anal. Calcd. (%) for C_15.5_H_24_CdN_8_O_7.5_: C 33.55, H 4.36, N 20.20; Found (%): C 33.25, H 4.22, N 20.30. FT-IR (ν, cm^−1^): 3129 (w), 2927 (w), 2870 (w), 1517 (m), 1447 (s), 1274 (s), 1132 (s), 990 (s), 903 (w), 858 (m), 817 (m), 738 (m), 657 (s).

### 3.2. Characterization Methods

Thermogravimetric analysis (TGA) coupled to differential scanning calorimetry (DSC) was carried out with a NETZSCH 449F3 instrument (NETZSCH TAURUS Instruments GmbH, Weimar, Germany) at a heating rate of 10 °C/min in a stream of helium, scanning range—30–750 °C. IR spectra were recorded on Agilent Cary 630 FT-IR spectrometer (Agilent Technologies, Santa Clara, CA, USA) equipped with a diamond ATR (attenuated total reflectance) tool (Agilent Technologies, Santa Clara, CA, USA). Elemental analyses were carried out on Carlo Erba CHNS analyser (Carlo Erba, Barcelona, Spain). The photoluminescence spectra were recorded on Horiba Jobin Yvon Fluorolog 3 Photoluminescence (HORIBA Jobin Yvon SAS, Edison, NJ, UAS) Spectrometer, equipped with 450 W xenon lamp, excitation/emission monochromator, and FL-1073 PMT detector. The powder X-ray diffraction data were obtained on a Shimadzu XRD 7000S powder diffractometer (Shimadzu Corporation, Kyoto, Japan) (Co-Kα radiation).

Continuous shape measures were calculated by the SHAPE 2.1 program [38] for seven-coordinated reference polyhedra [39].

### 3.3. X-ray Structure Determination

Single-crystal X-ray diffraction data were collected at 150 K (compound **2**) or 298 K (compounds **1** and **3**) on a Bruker-DUO APEX CCD diffractometer (Bruker Corporation, Billerica, MA, USA) (graphite monochromatized Mo Kα radiation, *λ* = 0.71073 Å, *φ* and *ω* scans of narrow frames) equipped with a 4K CCD area detector (Table 1). Absorption corrections were applied using the SADABS program [40]. The crystal structures were solved by direct methods and refined by full-matrix least-squares techniques with the use of the SHELXTL package [41] and Olex2 GUI [42]. Atomic thermal displacement parameters for non-hydrogen atoms were refined anisotropically. The positions of H atoms were calculated corresponding to their geometrical conditions and refined using the riding model.

## 4. Conclusions

In summary, coordination chemistry of a new ligand, 1,3-bis(1,2,4-triazol-1-yl)adamantane, was explored through the examples of silver(I) and cadmium(II) nitrates. Depending on the metal-to-ligand ratio, cadmium formed 1D coordination polymers with chains of different composition, while in the case of silver(I) ions, a 2D coordination polymer was obtained independently of the ratio of the reagents used. All three coordination polymers demonstrated ligand-centered emission.

## Data Availability

CCDC 2099260-2099262 contain the supplementary crystallographic data for this paper. These data can be obtained free of charge from The Cambridge Crystallographic Data Center at http://www.ccdc.cam.ac.uk/data_request/cif.

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
