# Peer review of "Synthesis, Crystal Structure, and Luminescence of Cadmium(II) and Silver(I) Coordination Polymers Based on 1,3-Bis(1,2,4-triazol-1-yl)adamantane"

_molecules, 2021, doi:10.3390/molecules26175400_

Round 1

Reviewer 1 Report

The paper of Potapov et al. reports on the synthesis and full characterization, also by X-ray diffraction methods, of new Ag(I) and Cd(II) coordination polymers containing the ditopic ligand 1,3-bis(1,2,4-triazol-1-yl)adamantane. The introduction is sound and comprehensive with the related literature well covered, the discussion of the experimental results is well done, clear and concise, and the structures of the compounds very interesting. In addition, a study on the optical properties of the compounds was rigorously carried out.

In summary I consider the paper suitable for publication in Molecules as it is.

Author Response

We are thankful for a high evaluation of our work.

Reviewer 2 Report

Authors prepared Cd(II) and Ag(I) coordination polymers containing 1,3-bis(1,2,4-triazol-1-yl)adamantine ligands. They well described crystal structures of Cd- and Ag-CPs, and analyzed luminescent properties. The article is well organized, but they did not study about biological activity even though they mentioned about it in Introduction. The article is recommended for publication on Molecules after major revision.

Revision points:

  1. The images of all Figures are not clear enough.
  2. In Figure 1, 3 and 5, they should add symmetric atom labels with symmetry operations.
  3. They mentioned that adamantanes and 1,2,4-triazole derivatives possess a wide range of biological activity. Have they tried to measure the biological activity for Cd- and Ag-CPs?

Author Response

Revision points:

1. The images of all Figures are not clear enough.

Unfortunately, figure quality was compromised during PDF conversion. All figures in the revised version were replaced by high resolution originals.

2. In Figure 1, 3 and 5, they should add symmetric atom labels with symmetry operations.

Atoms labels were added to the figures, symmetry operations were added to figure captions.

3. They mentioned that adamantanes and 1,2,4-triazole derivatives possess a wide range of biological activity. Have they tried to measure the biological activity for Cd- and Ag-CPs?

Biological activity, namely, the antibacterial activity of coordination polymers is indeed a very interesting area and it could be a topic of our future work. However, in our preliminary studies we have carried out some screening of the reported compounds for antibacterial activity on several lines of microorganisms, but no activity was observed probably due to the extremely low water solubility of these coordination polymers. We decided not to include these negative results in the current manuscript, since it was only a simple preliminary experiment.

Reviewer 3 Report

The article entitled “Synthesis, crystal structure and luminescence of cadmium(II) and silver(I) coordination polymers based on 1,3-bis(1,2,4-triazol-1-yl)adamantane” is focused on the synthesis and the characterization of three coordination polymers obtained by the reaction of the new ligand 1,3-(1,2,4-triazol-1-yl)adamantane with cadmium and silver ions. The structures of the new Ag(I) and Cd(II) coordination polymers was characterized by single-crystal X-ray diffraction analysis. The polymers were also characterized through powder XRD, thermal analysis (TGA-DSC), FT-IT spectroscopy and their luminescent properties were studied in the polycrystalline solid state.

The manuscript is well written and provides a deep analysis of the coordination polymers and their properties. I will recommend the acceptance after minor revisions (listed below).

Minor revisions:

  1. Please add 2/3 sentences to introduce the topic of the article.
  2. Cadmium(II) could be replaced by Cd(II) in various places, e.g., lines 88, 99, 106, 108, 113, and 115.
  3. Introduction, page 2 line 52. Modify as “spectrum of living organisms”.
  4. Introduction, page 2 line 45. Modify as “1,3-(1,2,4-triazol-1-yl)adamantane (L)”.
  5. The resolution of all figures in the main text should be improved.
  6. Figure 1. A second panel could be added to show the second position of the nitrate ion.
  7. Figure 2, 4, and 6. Please add labels for the key chemical components, to allow a better understanding of the structure of the coordination polymer.
  8. Results, page 4 line 128. Modify as “Powder X-ray diffraction (powder XRD),”.
  9. Results, page 5 line 131. Modify as “thermogravimetric analysis (TGA) coupled to differential scanning calorimetry (DSC).”
  10. Materials and methods, page 7 line 191. Change “resultant” in “resulting”.
  11. Materials and methods, page 7 lines 204-205. The heating rate is reported in K whereas the scanning range in °C, please use unify the temperature unit.
  12. Materials and methods, page 7 line 206. Change “FTIR” in “FT-IR”.
  13. Figure 2, 4, and 6. Please add labels for the key chemical components, to allow a better understanding of the structure of the coordination polymer.
  14. Supplementary Materials, Figure S2, S5, and S8. Please add labels for the key chemical components, to allow a better understanding of the structure of the coordination polymer.
  15. Supplementary Materials, Figure S4, and S7. Atom labels are not visible, please modify the figures to make them more readable.

Author Response

1. Please add 2/3 sentences to introduce the topic of the article.

The introduction was extended with some information on applications of coordination polymers (lines 48-52).

2. Cadmium(II) could be replaced by Cd(II) in various places, e.g., lines 88, 99, 106, 108, 113, and 115.

corrected

3. Introduction, page 2 line 52. Modify as “spectrum of living organisms”.

corrected

4. Introduction, page 2 line 45. Modify as “1,3-(1,2,4-triazol-1-yl)adamantane (L)”.

corrected

5. The resolution of all figures in the main text should be improved.

Unfortunately, figure quality was compromised during PDF conversion. All figures in the revised version were replaced by high resolution originals.

6. Figure 1. A second panel could be added to show the second position of the nitrate ion.

Figure showing both positions of nitrate ions was added to the Supplementary materials as Figure S1 (b).

7. Figure 2, 4, and 6. Please add labels for the key chemical components, to allow a better understanding of the structure of the coordination polymer.

Atom color legend was added to figure captions.

8. Results, page 4 line 128. Modify as “Powder X-ray diffraction (powder XRD),”.

corrected

9. Results, page 5 line 131. Modify as “thermogravimetric analysis (TGA) coupled to differential scanning calorimetry (DSC).”

corrected

10. Materials and methods, page 7 line 191. Change “resultant” in “resulting”.

corrected

11. Materials and methods, page 7 lines 204-205. The heating rate is reported in K whereas the scanning range in °C, please use unify the temperature unit.

corrected

12. Materials and methods, page 7 line 206. Change “FTIR” in “FT-IR”.

corrected

13. Figure 2, 4, and 6. Please add labels for the key chemical components, to allow a better understanding of the structure of the coordination polymer.

Atom color legend was added to figure captions.

14. Supplementary Materials, Figure S2, S5, and S8. Please add labels for the key chemical components, to allow a better understanding of the structure of the coordination polymer.

Atom color legend was added to figure captions.

15. Supplementary Materials, Figure S4, and S7. Atom labels are not visible, please modify the figures to make them more readable.

Figures S4 and S7 were modified, labels were enlarged and moved.

Round 2

Reviewer 2 Report

The Figure 1, 3 and 5 were not revised good enough. They should add symmetric atom labels with symmetry symbols and put symmetry operations in Figure captions. The example of addition of symmetric atom labels and Figure caption with symmetric operations is attached.

Author Response

Thank you for your suggestion, the labels and Figure captions were updated.